# Transferring Healthcare Professional’s Digital Competencies to the Workplace and Patients: A Pilot Study

**DOI:** 10.3390/ijerph192013187

**Published:** 2022-10-13

**Authors:** Olga Navarro Martínez, Jorge Igual García, Vicente Traver Salcedo

**Affiliations:** 1Department of Nursing, Catholic University of Valencia, 46007 Valencia, Spain; 2Instituto ITACA, Universitat Politècnica de València, 46022 Valencia, Spain; 3Instituto de Telecomunicaciones y Aplicaciones Multimedia (ITEAM), Departamento de Comunicaciones, Universitat Politècnica de València, 46022 Valencia, Spain

**Keywords:** digital health, digital competencies, healthcare professionals, e-learning, healthcare education

## Abstract

The new times, marked by immediacy, globalization, and technological advances, has forced health professionals to develop new competencies to adapt to the new challenges. However, necessary skills such as using digital tools are primarily ignored by institutions, hospitals, and universities, forcing professionals to undertake training in these areas independently. This research aims to analyse if there is a transfer of what has been learned in the digital healthcare field to their professional practice and patients. To perform the study, 104 healthcare professionals, mostly nurses, who had completed online training in digital competencies answered a questionnaire with 17 questions. These questions were related to the transfer of learning to professional practice and its use for developing patient resources. Almost 60% of the professionals said that they have used what they learned in the course in their professional work, but only 16% of the participants use it daily. The main barrier to not having applied what was learned during the course, according to participants, was the situation experienced during the COVID-19 pandemic, followed by a lack of time and lack of resources. Only 23 people out of 104 developed patient resources after the course; the most created were infographics and videos. In addition, 38 people used what they learned to improve their personal productivity: searches, storage, calendars, etc. Only 11 used it for research purposes. People between 31 and 40 years old create the most patient resources and use what they learn most frequently. There is a need to improve e-learning to provide quality training that can transfer good behaviour to professional practice in the health field.

## 1. Introduction

Many people use technology daily to study, work, have fun, and interact with others. This technology is also a valuable resource in the field of health. In recent decades, much effort has been invested in this area in devices that facilitate monitoring, diagnosis, treatment, and even the care of patients [1]. The use of digital media by health professionals for all the above-mentioned purposes implies a proactive and positive attitude and education and training. However, it is important to remember that 43% of the European adult population does not have sufficient digital skills to study, interact, or work in digital environments [2].

Few healthcare professionals generate digital resources such as videos, blogs, infographics, and podcasts [3]. However, they are powerful tools for the follow-up of people with chronic pathologies [4], which can be a privileged scenario to understand better the needs and interests of people. We will be able to use these tools to understand their health-related behaviour [5,6]. Moreover, we can use all this knowledge to educate the population and offer quality healthy advice [7]. Therefore, the health information found in the network is at risk of being erroneous and unverified [8,9,10,11]. Among the possible reasons professionals do not develop their resources is due to a lack of time [12] but also a lack of skills and digital competencies [13,14,15,16]. The limited institutional support and low training professionals receive make this gap even more complex between the popular demand and what healthcare workers offer [17].

It is, therefore, important to find out whether health professionals apply what they have learned in their work after receiving specific training in the use of the technology. This point is essential because, as Cuadrado et al. pointed out, “one of the main purposes of any training proposal is to enable and facilitate the transfer of learning to professional practice, thereby avoiding the gap between the world of work and the academic world” [18]. However, the training that health professionals receive, especially in digital environments, is based solely on providing information in the form of theoretical content. Generally, this training is followed by a satisfaction survey and a test [19,20]. This online training system falls far short of the expected practical transfer and is only at the initial levels of the model proposed by Kirkpatrick [21,22]. Kirkpatrick [23] described a method for evaluating training programs globally based on four levels of development, which are represented in the form of a pyramid. Figure 1 shows the graphic representation of this model. This model for evaluating training schemes includes four levels: level 1 (reactions) assesses student satisfaction with the training received; level 2 (learning) evaluates knowledge by employing questionnaires, exercises, tests, or skills in the form of practical activities or simulations; level 3 (behaviour) evaluates putting knowledge into practice and the existence of a behaviour change; in this case, it would involve the use of digital resources in their professional daily practice. Finally, level 4 (results) assesses the impact of these schemes on patients’ health.

It could, therefore, be considered that the ultimate purpose of any training in the healthcare field is, in a nutshell, to reach level 4, in other words, to bring about improvements in people’s health. To bring these aforementioned improvements to processes, techniques, procedures, and all kinds of tasks that are carried out with patients, it is essential that level 3, applying what has been learned to professional practice, is achieved. Therefore, level 3 is the only one directly related to the achievement of results and can be considered transfer criteria and should, therefore, be included as a requirement in healthcare training [24]. Nevertheless, and even though parameters such as satisfaction (level 1) and knowledge (level 2) are often used to assess training suitability, some authors have pointed out that the knowledge and skills learned by nurses are often sufficient to motivate the transfer to practice without reaching level 3 or higher [25].

Studies related to transfer have been carried out nowadays, especially among newly graduated professionals or students [26,27,28]. However, the purpose of this study is to analyse whether, after receiving e-health or e-skills training, the professional applies it to their work, both for their own performance and for the benefit of patients. Therefore, the objective of this exploratory study is to analyse, utilising consultation with healthcare professionals, if there is a certain transfer of what has been learned in the digital healthcare field to their professional practice and patients. Once they have acquired certain digital skills or competencies, are they applied in their professional practice? How often? What are they used for? What is to stop them from using these resources more?

It is also of interest to analyse whether there are differences between certain professional categories, and which professionals are those who use them most once trained. Does the age of the professional or his/her workplace have an influence?

In this article, we talk about health professionals, which encompass all types of profiles related to patient care and attention: nutritionists, physiotherapists, doctors, nurses, technicians, etc.

## 2. Materials and Methods

To implement this study, a questionnaire was developed ourselves for exploratory purposes among healthcare professionals who had already received training in digital competencies through courses offered in an online environment. For this purpose, the collaboration of two specialised online training institutions that included this theme in their catalogues was requested to send the survey to the graduates of their courses. These two companies offer online training in digital skills for all types of healthcare professionals: doctors, nurses, physiotherapists, etc. The type of training generally consisted of watching videos and multimedia materials, followed by a test-type exam. The survey was sent to all students who had already completed one of these courses, inviting them to participate in the study. 

The students carried out the training courses on their own initiative and paid for them themselves.

Due to the difficulty of locating health professionals who have already been trained online in the use of digital tools and refining the questionnaire, and adapting it to future, more extensive research, the sample of participants was a convenience sample. The purpose of using this type of sampling was to make a first approximation to the research on the transfer of digital competencies to the work environment in the field of health. 

Following approval by the collaborating companies, a survey of our elaboration was carried out that included 17 questions in 5 sections or blocks. The questionnaire was sent to 1000 people who had already undergone training on this topic.

### 2.1. Structure and Details of the Survey 

As in the first step, the questionnaire was hosted on Google Forms and could be answered online from any device. The answers in this case were automatically stored for further processing. It was decided not to collect the participants’ email addresses, for data protection reasons.

On the first screen, the participants accessed a brief explanation of the survey’s objective, the ethics committee’s approval, and the responsible persons’ data. They also had to press the “I accept” button as consent before starting the survey itself. The estimated time to answer the survey was less than 10 min.

The standard questionnaire was structured as follows: The first five questions referred to the training received (number of hours, was this time sufficient, have you done more courses of this type, etc.?). The following five were related to demographic data. The following two questions referred to the general application of what you have learned, i.e., how long ago did you finish the training? Do you use what you have learned during the course? Depending on the answer to this last question, we proceed to the last block to be answered, corresponding to the use or non-use of what has been learnt. The professional answers four more questions if the participant has used what they learned in the course. Questions 12 to 16 evaluated the transfer (level 3 of the Kirkpatrick model). If the participant has not used anything, the professional answers only one question about the reasons.

These items were grouped into five sections where the participants could change their answers if they wished before finally submitting them. The questions were mandatory except for those in blocks 4 and 5, which were optional depending on the answer given in the third block.

The collaborating companies sent the link to the survey to the graduates who were on their database. No registration was required to reply, and the personal data requested were of a general nature (professional field, job profile, and age at intervals). No consideration was given to the “gender” variable in this research. As it was of particular interest to find out whether or not transfer occurred after learning, it was not considered essential in this case to differentiate between men and women. The survey was actively receiving responses from February to December 2021. The complete survey is in Appendix A.

The ethical approval to carry out this survey was obtained from the Research Ethics Board at the Polytechnic University of Valencia, Spain (P4_25_07_18). Participation was free and entirely voluntary.

### 2.2. Statistical Analysis

Different test statistics were applied to check if the results were statistically significant (*p* < 0.05). Most variables were categorical, and some were dichotomous. We used the chi-square as the test statistic with the corresponding degrees of freedom depending on the dimensions of the contingency table. Some variables, such as age, were considered ordered variables, grouped in different age intervals. In those cases, we used the Kruskal–Wallis test and correlation.

## 3. Results

The questionnaire was sent to 1000 students and the response rate was approximately 10%. Two reminders were sent during the time the questionnaire was open in order to increase the number of responses. Finally, 104 responses were received from people who had taken courses in one of the two collaborating companies. Of these, 37 responses were received from nurses, representing the majority group, followed by the group of nursing assistants with 24 responses, and then laboratory technicians with 16 responses; other occupational profiles are detailed in Figure 2.

Participants in this survey carry out their professional work mainly in hospitals, with 65 people marking this option as a workplace, 21 indicating that they work in primary care, 6 work in residences for the elderly, 3 work outside the hospital, 3 work in professional mutual societies, 3 do not currently work, 2 of them are currently working at university, and, finally, one of the professionals works in the private sector. All participants work in Spain.

The age distribution of participants is 1 person under 20 years of age, 26 people from 21 to 30 years old, 23 from 31 to 40 years old, 42 belong to the group from 41 to 50 years old, and the last group, from 51 to 60 years old, has 12 people. In the 51–60 group, 75% are nursing assistants.

As for the number of hours performed, they range from 22 h related to a short course to 1500 h of completion of a master’s degree, with the majority of participants having completed courses of 80 h (67 participants) followed by 22 h training courses (14 participants). Only 19.23% (20 people) have completed a training course of over 80 h.

If we analyse the number of hours of training performed by age groups, we obtain the fact that the average number of hours is higher in the group of 31–40 years of age, with 187 h of training per person. Details of this information are displayed in Table 1.

If we analyse only the hours of training carried out by the nurses, we conclude that, similarly, the age group where more hours have been carried out for training in this area is that of 31–40 years of age (12 people), with an average of 270.36 h per person.

A proportion of 87.5% of the participants were satisfied with the hours received during their training. Of the 104 participants, 73 (70.19%) had never completed another course on digital competence or eHealth previously. Twelve participants had once conducted a course on this subject, ten had completed two courses, and nine people had completed more than two training programs in addition to the recently completed course.

Regarding the time that had elapsed since the training on which it was consulted, there was also a significant variability, going from a period of less than one month to more than nine months from completion. The group of 6 to 9 months was the least represented with 8 responses, and the group that was most represented was the group of over nine months with 29.

### 3.1. With No Transfer to the Workplace

Regarding the question “Have you applied any of the things that learned?”, 62 people (59.61%) answered yes, while 42 (40.39%) indicated that they had not done so. All participating physiotherapists in the survey answered yes to this question (5 people). Table 2 shows the number of each profile that has or has not used what they have learnt during the course.

As for the age of participants, we observe that the 41–50 age group is the group that has stopped applying what has been learned with 33.33%, whereas the other groups exceed 40% (21–30 years, 42%; 31–40 years, 47%; 51–60 years, 50%).

If we relate the non-application of what we learned to the time that passed since the training period finished, there is no happy medium, i.e., the period of less than one month since the end of the course and that of more than nine months after the completion are those that present a higher percentage of people who have not applied what they learned. In Table 3, the results obtained in this comparison can be observed.

A proportion of 41.53% of people working in a hospital environment and 42.85% of those who do so in primary care have not applied what has been learned, and there are no significant differences between the two professional areas.

As for the barriers or difficulties involved in not applying what has been learned in their course to their professional practice, the main reason, according to the respondents, is the current situation (COVID-19 pandemic) being pointed out by 18 people (42.85%), followed by a lack of time (8 people, 19.04%), a lack of resources (8 people, 19.04%), a lack of institutional support (6 people, 14.28%), a need to further their training (6 people, 14.28%), non-receptive patients (5 people, 11.90%), and a lack of motivation (3 people, 7.14%). Seven of the participants did not choose any of the options proposed. It is interesting to point out that six respondents selected the option “I plan to do so later”. In this question, it was possible to indicate several options. A total of 100% of the people who pointed out the lack of institutional support as an obstacle were nurses, 75% selected the lack of time or resources, 66% reported a lack of motivation, and 50% indicated that they needed further training.

### 3.2. Positive Transfer

In terms of people who have used what they learned in their course (62 people), it is noted that 46.77% often apply it, 37.09% use it sometimes, and 16.12% apply it every day. We found a strong dependency (*p* = 0.0238) between the duration of the course and its use: the longer the training course, the more it is used; for example, while no one attending short-term courses (less than 22 h) uses it every day, the percentage increases to 10.8% for mid-term courses and 42.8% for long-term courses (more than 150 h). There are also differences between the use and the profession (*p* = 0.0501): only 12.5% of nurses use what they learned in that course daily compared to 75% of doctors surveyed, although 45.83% of nurses claimed to use it frequently. In addition, 80% of physiotherapists use what they learned in the course only sometimes, as opposed to 33% of nursing assistants. A proportion of 50% of nursing assistants use it frequently.

If a comparison is made between different age groups, we can see that the 51–60-years-old group is the one who most uses what they learned in the course “often” at 66.66%, followed by the 21–30 age group at 60%, the 31–40 age group at 41.66%, and the 41–50 age group at 39.28%. However, if we compare different age groups as for those who apply it every day, we can conclude that it is the 31–40 age group at 33.33% who put it into practice most on a daily basis, while from the 21–30 group, only 6.66% apply it every day, and the group of 51–60 years old never use it daily. A proportion of 42.85% of the participants aged 41–50 apply what they have learned only at times. These comparisons can be observed in detail in Figure 3 (*p* = 0.4061).

To determine what these resources have been used for, question 14 offered different options, and several possibilities could be marked. Thirty-eight people indicated that they had used it for their productivity, 23 used it to develop patient resources, sixteen participants used it to improve their brand and communication on social networks, and 11 used it for research purposes. Each person could pick several answers.

As for nurses, 62.5% admitted using what they learned in their training to improve their personal productivity, 45.83% specifically to elaborate resources for patients, 29.16% to build their personal brand and communication on social media, and 25% for research purposes. All doctors claimed to use what they learned to make resources for patients and improve their productivity; none used it for research purposes, and only two used it to enhance their brand and communicate on social networks. Figure 4 shows the comparison of the percentage of use of what was learned among doctors, nurses, and all other profiles.

The age group that developed patient resources the most was the 31–40 age group (58.33%), followed by the 41–50 group with 35.71% and the 21–30 age group with 33.33%. The 51–60 group did not report developing patient resources.

Finally, they were asked what type of resource had been developed and to choose from a list that included blogs, podcasts, videos, infographics, profiles on social networks, and others, and how many of these resources were created. Fifteen participants indicated having made infographics, 14 stated they had created videos, 8 had created profiles on social networks, 4 created blogs, and only 3 made podcasts; 33 of the respondents did not indicate having made any resources. 

It is interesting to highlight that 75% of the people who did more than 80 h of training in courses applied what they had learned in their training as opposed to 55.95% of people who completed less than 80 h of training (*p* = 0.0115). In addition, 58.33% of people who performed more than 80 h of training belonged to the 31–40 age group, but no statistical difference was found between course duration and age groups (*p* = 0. 6072).

## 4. Discussion

In this pilot exploratory study, the responses of healthcare professionals who have conducted online training about digital healthcare were recorded. For this purpose, this survey was carried out in which different professional profiles participated.

After analysing the data obtained in this survey, the initially assumed hypothesis can be confirmed as 60% of the professionals who have been trained in digital skills in the healthcare field apply their knowledge and transfer it to their professional practice. Logically, the number of hours of training would influence this transfer; according to the analysed results, we can affirm that the greater the number of hours of training, the more likely one is to apply what has been learned to practice. However, most participants chose 80 h training options or shorter, probably by having little time to spend on training. It is important to note here that the work overload perceived by professionals is an obstacle to the success of continuing training [12].

According to the results obtained, age also seems to be a key factor. More than 50% of professionals over 30 use what they learned in the course every day compared to 6.66% of the youngest professionals. These data could reaffirm what other studies have suggested; nurses and younger healthcare professionals use fewer digital resources with their patients [3,29]. As reported by these studies, this could be due, in the case of nurses, to a generation gap or to a lack of training received by younger nurses during their undergraduate studies. However, as they themselves pointed out, more research is needed to be able to conclude the causes of this age disparity. This group has developed the lowest number of resources for patients, only ahead of the 51–60-year-old group that did not develop any resources.

The most trained health professionals are those from 31 to 40 years old; they are also the group that most uses what they learned daily and the one that has created the most resources. This group of professionals is similar to the group that has created the most resources for patients and may consider this indicator a true transfer criterion according to the Kirkpatrick model [23]. However, it is essential to point out that this age group is also the one that has carried out the most extended training sessions (58.33%).

There is no relationship between the workplace and the application of what was learned during the formation.

The time elapsed since the completion of training also appears to be a factor influencing the transfer of what has been learned to professional practice because, as we have observed, people who have recently completed training and those who have been doing it for longer tend to use less of what they have learned. However, it is essential to remember that the change in practice is not limited and may occur sometime after the educational intervention [25]. Six respondents indicated that they intended to apply it later; in these cases, the intention is strongly linked to a real change in behaviour [25]; therefore, these people could be expected to apply what they learned in the future.

Regarding the analysis of causes for which a professional has not applied what was learned in training, the most prominent cause for all professionals not applying what has been learned is the current situation, referring to the current pandemic of COVID-19. Given the large number of human resources that were required to deal with the pandemic, it is understandable that this argument is cited as a major cause for the non-utilisation of what was learned. However, it would have been just during this period where it could have been more interesting to create and share resources with patients and carry out online follow-ups [30,31,32,33,34]. It is of particular interest to point out that the case of “lack of institutional support” is only referred to by nurses. Davids [35] pointed out that motivation to learn and to carry out this transfer are not always directly related to the training received but rather other factors such as work climate and institutional support. To consolidate learning, opportunities are needed to apply what has been learned in the professional environment [24]. This is where healthcare institutions play an essential role [12].

As for the use that professionals make of what they learned in the course, it is interesting to highlight that the majority of respondents have applied it to improve their productivity, networking, and research, which would correspond to the first two levels of digital competence according to the Digcomp framework [36], information and literacy, and communication and collaboration. However, very few professionals have developed patient resources, which would mean reaching a higher level within that framework of competencies. This could be due to the design of the training itself; for this reason, it would be interesting if online training in digital healthcare ensures that the knowledge generated can be transferred into practice and influence sustained behavioural change [22]. 

The most developed resources by the participants were infographics and videos, two very useful resources to educate in healthcare and help patients self-manage themselves better [37,38,39]. Infographics could be even better accepted by patients than videos [38] and are a powerful resource for sharing knowledge among healthcare professionals [40] and spreading research results [41].

## 5. Limitations

As for limitations, it can be noted that the contents and activities included in each training session were not taken into account, only the hours that the courses lasted, but the content index was verified to confirm that issues on health and digital skills were studied. However, the people who had already carried out the training based on motivation or interest in the subject did not all apply what they learned later. The limited sample makes it complex to generalise the results obtained when conveyed to the population; however, it could be an initial approach to future studies in this area. The representativeness of specific occupational profiles or age groups may be biased in the results obtained. In terms of the suitability of evaluating the transfer or behaviour (level 3 of the Kirkpatrick model) by employing a survey, there are studies suggesting that a post-training survey is a valid measure of transfer to practice as it provides information on the intention to change the practice to improve [25]. However, it would be advisable to use other complementary evaluation measures such as the consultation of co-workers and supervisors or managers [35].

## 6. Conclusions

Evaluating the knowledge transfer of a health professional to their professional practice after digital health training is not easy. However, it is essential to measure this parameter to determine the learning impact and, thus, estimate health outcomes. Training companies, universities, and health institutions should develop strategies to evaluate that transfer as an indication of the quality of the instruction received.

In this study, professionals over 30 years of age are those that are primarily formed in this theme and also those who most apply what they learned daily, although they continue to develop resources for patients. These professionals could pave the way for younger colleagues to achieve an increased transfer impact. 

Although there are many barriers to the applicability, the situation caused by the COVID-19 pandemic has meant a considerable saturation in healthcare professionals, which may have diminished their professional growth in other areas such as eHealth or innovation. It is necessary to design spaces for collaboration and learning, provide resources, and allow adequate time for practical implementation in the working environment, with the support of organisations, so that the effort made by the professionals in training shall not be in vain.

To determine if nurses are the least trained, whether they apply what they have learnt daily, and whether they generate fewer resources for patients than other professionals, it is necessary to continue researching on these lines. Nevertheless, it is rewarding to assess these conditions to implement measures that reduce this gap that could bring safe nursing practice for patients and society and affect nurses’ professional development.

## Figures and Tables

**Figure 1 ijerph-19-13187-f001:**
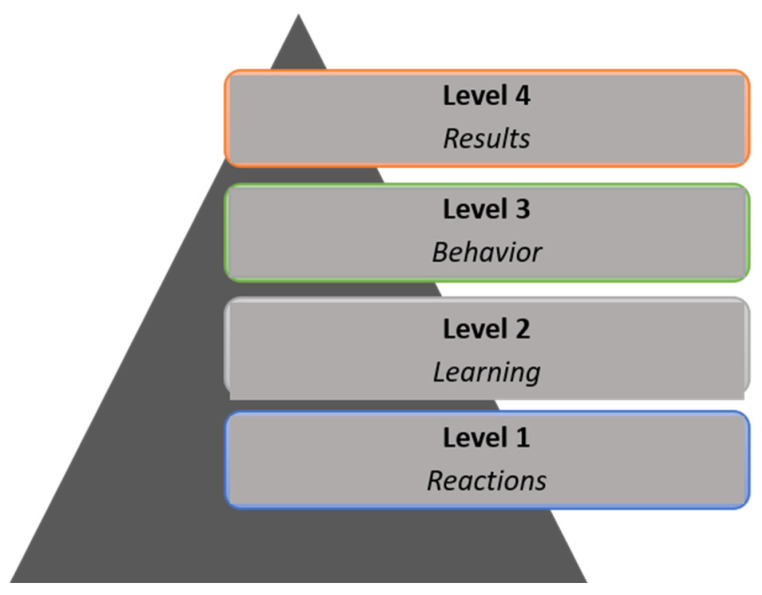
Kirkpatrick model.

**Figure 2 ijerph-19-13187-f002:**
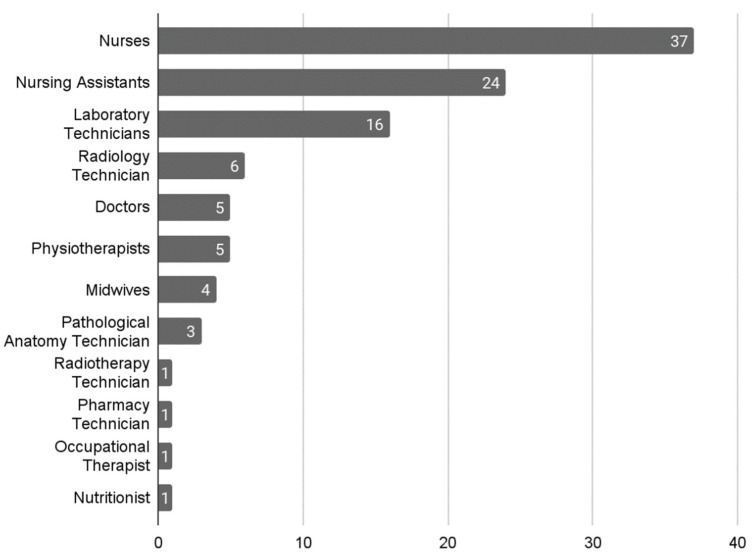
Distribution according to professional profiles.

**Figure 3 ijerph-19-13187-f003:**
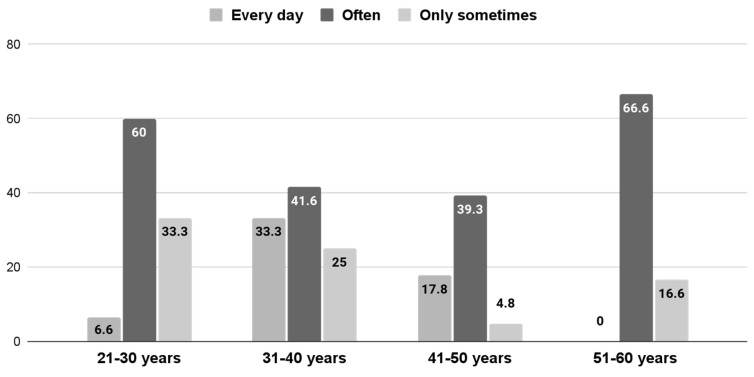
Frequency of use by age group in %.

**Figure 4 ijerph-19-13187-f004:**
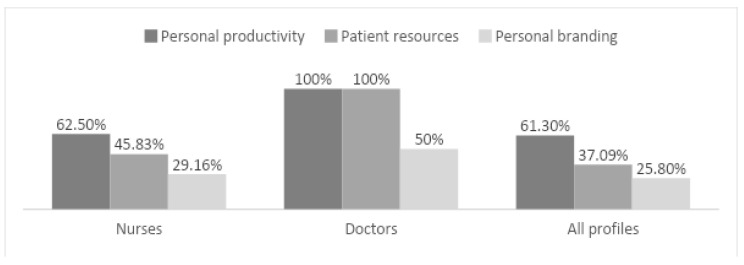
Comparative use of lessons learned by profiles in %.

**Table 1 ijerph-19-13187-t001:** Hours of training by age group.

	20 or Less	21–30	31–40	41–50	51–60
Total training hours	150	1916	4304	5094	1086
Average training hours	150	73.69	187.13	121.28	90.5

**Table 2 ijerph-19-13187-t002:** Application and non-application of what has been learnt by professional profiles.

	Have You Applied Any of the Things That Learned?	
	Yes	No
Nurses	24	13
Nurses assistants	12	12
Laboratory technicians	9	11
Doctors	4	1
Nutritionist	1	0
Midwives	2	2
Radiology technicians	3	0
Physiotherapists	5	0
Other technicians	2	2
Occupational therapists	0	1
Total	62	42

**Table 3 ijerph-19-13187-t003:** Failure to apply lessons learned and time elapsed.

Time That Has Elapsed since the End of the Training Period	<1 Month	1–2 Months	3–6 Months	6–9 Months	Over 9 Months
Number of persons who have not applied what has been learned	11	7	6	2	16
% of people out of total over that period	50%	30.43%	27.27%	25%	55%

## Data Availability

Not applicable.

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
