# Peer review of "Transferring Healthcare Professional’s Digital Competencies to the Workplace and Patients: A Pilot Study"

_ijerph, 2022, doi:10.3390/ijerph192013187_

Round 1
Reviewer 1 Report
I liked your paper. I think it is useful for the topic of digital training for healthcare staff, and will be an interesting addition (though not earth-shattering new knowledge) to the body of knowledge about training in digital skills for use in practice. I do however feel that it needs some tidying up to do justice to your study and for clarity for the readers. If this is done I would be happy to see it published.
I have put my detailed comments and suggestions into a Word document, attached.

Author Response
REVISOR 1
Dear Reviewer:
We really appreciate your comments. Below, we are detailing how we have approached your suggestions that have help us to increase the quality of the paper.
I have the following detailed comments and suggestions:
- L38-43 This is a very long sentence; splitting it up will make it more understandable and easier to read. Changes made following your suggestion. Now, it’s easier to read and more understandable
- L44 I was unsure what you meant by 'health network' until I looked at the references. Perhaps put 'health information online' instead. Changes done in order to avoid confusion following your suggestion.
I like the use of the Kirkpatrick model.
- L89 I would be happier if you put his/her instead Changes made
You have clearly stated your hypothesis and the objectives can also be found reasonably clearly.
- Did you pilot your pilot survey with just a few people? If not, why not? It is usually a useful exercise to tweak questions (see comment about questions).
In fact, this pilot is just for that purpose, to "test" the survey to be able to do a subsequent study with a larger volume of people. It is not easy to find, at least in our environment, health professionals who have already undergone specific training on digital skills. This was the major goal of our pilot.
- Either write what your survey questions are in full in the body of the paper, or
signpost to the appendix, which I had to look for myself as I was not directed to it in the text.
Line 136 is indicating that the whole questionnaire was found at the end as an annexe.
- Questions 8 and 10 are pretty similar: what was the reasoning behind the wording?
Sorry, as it was probably a translation error. Question 8 refers to the country in which the professional works and question 10 was intended to find out where the professional works, i.e. does he/she work in a hospital? in a health centre? We have modified question 10 to make it easier to understand.
- L151 ‘that laboratory technicians....’ Should be ‘then laboratory technicians...’ Changes made
- L164-168 could be clearer. L165 ‘at’ should be ‘to’ and ‘being’ shouldn’t be there I
think. Revised and changed
- Table 1 needs to be clearer: what do you mean by ‘half of hours of training’? Do you
mean ‘median’? % of total: which total? We really appreciate your inputs in order to make clearer the paper. Indeed, this table refers to the average number of hours of training completed by professionals. We have removed the last row (% of total) to avoid confusion.
L187-191 It would be good to have a table of this information with % of use for the numbers of participants in each professional category. I’m not sure that comparing the percentage knowledge use between the different professional categories is useful as the numbers of people in some categories is so small and some much higher (5 doctors compared with 37 nurses?). You mention this later in your discussion (L339-40)
The table is drawn up according to suggestions, and the percentages have been completed with absolute numbers to avoid confusion.
- Why have you put the decimal point as a comma in the tables and as a point in the
text? It would be good to be consistent (and I would use a point) Changes made following your suggestion.
L214-216. L218-223: same comment as for L187-191
- L237 ‘sixteen’ Changes made. Thanks for detecting such issues.
- L263-274 Sentence doesn’t make sense. I might write ‘This pilot exploratory study
probed the reponses....’ This part has been removed to avoid confusion.
- L278-282 Also logically if more hours were spent training are staff more likely to
apply their training to practice especially if they were funding the training
themselves, to justify their outlay if more hours equalled more expense?
You are probably right, but we do not currently have the data to confirm this hypothesis. It would be interesting to analyse whether, given the same price, professionals would choose longer courses or whether they would have continued to opt for shorter options. It is very interesting to think that if I spend more money, this "obliges" or predisposes me to use more of what I have learnt.
- L309-310 I agree that during the pandemic would have been an ideal time to use all
these skills sharing digital resources, but I think if Spain were anything like the UK,
creating the resources would have been just too much work on top of that of the
already/almost burnt-out population of healthcare professionals. You do mention this point in
L357 but in that sentence I think you mean that healthcare professionals
have been saturated with work
Unfortunately, the situation in Spain during the pandemic was most likely similar to that experienced in the UK. Health professionals were overwhelmed by the workload they had to cope with, which most likely made it difficult to apply what they had learned. We qualify that part to clarify that we think overload was responsible for this shortcoming.
My thought: if there was some sort of financial help to pay for this training with the requirement that it was put into practice use, would this not increase the transfer of knowledge to practice?
It is an exciting proposal, but would "forcing" professionals to apply what they have learned have the expected effect? We keep working on this area, and such a hypothesis is worthy at least to be explored.

Reviewer 2 Report
In order to adapt to the digital age, health professionals should meet the requirements of the times and improve their digital application practice capabilities. The study focuses on whether health professionals can apply digital domain expertise to work practices and patient care processes, and analyzes the barriers to health professionals learning digital domain expertise, and assesses the feasibility of applying digital domain expertise to professional practice. This study requires multi-agent collaboration at the macro level to promote the practice of digital technology application in the health field.
Comment 1
The abstract describes the necessity, methods, questionnaire data, and countermeasures for health professionals to apply digital skills to professional practice. However, the text description stays on the analysis of the absolute value of the survey, but the introduction of the highlights is not prominent, and the introduction of the results is relatively simple. It is recommended to describe the highlights of the study.
Comment 2
Line 94-95
The description of the selection procedure of the survey samples is relatively brief, and it is recommended to add more details. Is the study randomly selected? Or stratified sampling? The number of persons in different categories of the survey samples is quite different, and it is suggested to explain the reasons.
Comment 3
Line 134-135
The questionnaire design did not take into account variables such as gender and education. It is suggested to supplement the description of the inclusion of the above variables.
Comment 4
Line 149-150
The authors did not state the recovery rate of the questionnaire and the effectiveness of the questionnaire. It is recommended to supplement the above.
Comment 6
Line 227-232
The practice of digital technology in different age stages is described, reflecting the gradual decrease of age groups 51-60, 21-30, 31-40, 41-50, and 21-30. However, the reason is not explained later, and it is recommended to add.
Comment 7
Line 307-309
This study describes the barriers to the adoption of digital technologies in the health sector, mainly due to the limitations of the COVID-19 pandemic. However, it did not take into account the support of the venue's hardware facilities and policy background. Please describe the effect of the policy on the application of digital technologies.
Author Response
REVISOR 2
In order to adapt to the digital age, health professionals should meet the requirements of the times and improve their digital application practice capabilities. The study focuses on whether health professionals can apply digital domain expertise to work practices and patient care processes, and analyzes the barriers to health professionals learning digital domain expertise, and assesses the feasibility of applying digital domain expertise to professional practice. This study requires multi-agent collaboration at the macro level to promote the practice of digital technology application in the health field.
Thank you very much for your input and suggestions. We hope to have improved the article based on your review.
Comment 1
The abstract describes the necessity, methods, questionnaire data, and countermeasures for health professionals to apply digital skills to professional practice. However, the text description stays on the analysis of the absolute value of the survey, but the introduction of the highlights is not prominent, and the introduction of the results is relatively simple. It is recommended to describe the highlights of the study.
The abstract has been modified according to the suggestions received from your side and other reviewers. Now, it is much more evident the added value the paper is bringing..
Comment 2
Line 94-95
The description of the selection procedure of the survey samples is relatively brief, and it is recommended to add more details. Is the study randomly selected? Or stratified sampling? The number of persons in different categories of the survey samples is quite different, and it is suggested to explain the reasons.
As indicated in the text, the sample chosen in this case is a convenience sample, as this was the initial stage of this research. This pilot study aimed to assess this questionnaire's use and obtain an initial approximation of transfer. It was not randomised as it was sent to people we knew had done training. However, we did not initially know whether or not these people had applied what they had learned. In the future, once more resources have been secured, a larger-scale survey will likely be carried out, taking into account stratification to ensure that all health professions are represented. Several sentences are added, which hopefully clarify the followed procedure.
Comment 3
Line 134-135
The questionnaire design did not take into account variables such as gender and education. It is suggested to supplement the description of the inclusion of the above variables.
The main objective of this study was to make a first approach to a poorly -studied topic. We were particularly interested in finding out whether or not transfer occurred after learning. Since the sample was limited, we did not consider it essential to differentiate between men and women. Nevertheless, it is a valuable recommendation to be considered for a larger-scale study. A short explanatory sentence has been added to the text around this topic.
As for the level of education or previous training, the text does indicate that 73 people had never taken a course on this topic before.
Comment 4
Line 149-150
The authors did not state the recovery rate of the questionnaire and the effectiveness of the questionnaire. It is recommended to supplement the above.
An explanation about this issue is added in the text.
Comment 6
Line 227-232
The practice of digital technology in different age stages is described, reflecting the gradual decrease of age groups 51-60, 21-30, 31-40, 41-50, and 21-30. However, the reason is not explained later, and it is recommended to add.
An explanation about this topic has been added in the text.
Comment 7
Line 307-309
This study describes the barriers to the adoption of digital technologies in the health sector, mainly due to the limitations of the COVID-19 pandemic. However, it did not take into account the support of the venue's hardware facilities and policy background. Please describe the effect of the policy on the application of digital technologies.
A detailed editorial review of the paper has been performed to avoid confusion based on this comment. The COVID-19 pandemic introduced some limitations in our study but the barriers detected are not only due to the COVID-19 pandemic. Some policy strategies have boosted the adoption of digital technologies in the healthcare field, although the long-term effect is still being analysed. In any case, and we have reinforced such an idea in the whole paper, we have focused on how digital health learning outcomes after training are really applied in this field.

Reviewer 3 Report
Major issues
1) Hypothesis: You wrote in lines 90-92 that “the hypothesis … professional practice.” Why is this a hypothesis? Hypothesis should be based the existing literature, which I cannot find in your paper.
2) Regarding the first comment, are there any existing studies that examine any transfer of knowledge to their professions? Specifically, is the focus of your paper transfer of knowledge that is obtained digitally? Or the transfer itself? I can easily assume that that must be existing literature about transfer of knowledge to their professions in non-digital setting. If so, why do we care about transfer of digitally learning? Specifically, what is the theoretical contribution?
3) In accessing the professional’s digital competencies, did you control their previous experience on digital training? Differences in absorptive capability matter. If you were not able to obtain this info, this might be useful in the future study.
4) Lines 288-290: This data could reaffirm what other studies suggested; nurses and younger healthcare professionals use fewer digital resources with their patients p-[26.27]. -> What is the rationale behind this? What do the existing study say?
5) Figure 5: Why do we care about the number of resources created? The more the better?
6) Similar to the comment 5, what do we care about different type of resources that have been developed? Theoretically, or at least practically, what is the key difference among blogs, podcasts, videos, infographics, profiles, etc.? You should have at least provided the evidence that some type is more efficient in communicating with patients or transferring knowledge.
7) Isn’t there any need to differ the transfer to the workplace and patients? I believe the content will be different.
Minor issues
1) The limited institutional support and low training professionals receive make this gap even more complex between the popular demand and what healthcare workers offer [17]. -> This sentence can be just added to the previous paragraph, not requiring a separate paragraph.
2) Nevertheless, and even though parameters such as satisfaction (level 1) and knowledge (level 2) are often used to assess training suitability, some authors point out that the knowledge and skills learned by nurses are usually transferred to their practice [26]. -> Don’t quite understand why this is problematic. You said, transferring of knowledge is important, right? This needs more explanation.
3) Line 237: six teen -> sixteen
4) Line 239: tick -> pick
5) According to the professional profiles there are only 5 doctors out of 104 respondents, and most of them are nurses. This can be problematic as people generally think of doctors when they say health professionals. That said, isn’t it the case that the program itself is more useful for nurses than doctors?
Author Response
REVISOR 3
1) Hypothesis: You wrote in lines 90-92 that “the hypothesis … professional practice.” Why is this a hypothesis? Hypothesis should be based the existing literature, which I cannot find in your paper.
The reference to a hypothesis has been eliminated to avoid confusion since, as indicated, the article's aim is only exploratory to establish the beginning of new, more complete and in-depth studies on this subject.
2) Regarding the first comment, are there any existing studies that examine any transfer of knowledge to their professions? Specifically, is the focus of your paper transfer of knowledge that is obtained digitally? Or the transfer itself? I can easily assume that that must be existing literature about transfer of knowledge to their professions in non-digital setting. If so, why do we care about transfer of digitally learning? Specifically, what is the theoretical contribution?
Indeed, there are many studies on the transfer of learning to the work environment. Thorndike already described this learning theory in 1901 as "transfer of practice". Schunk in 1996 defined transfer as a process that refers to applying what has been learned in new contexts, or in new ways. In the 1980s, Benner did a lot of research on the transfer of learning to professional practice, especially in nurses and doctors. Studies related to transfer have also been carried out nowadays, especially among newly graduated professionals
(Botma, 2016) https://www.sciedupress.com/journal/index.php/jnep/article/view/8777/5989
(Elliot, 2020) https://pubmed.ncbi.nlm.nih.gov/31777092/
(Forbes, 2018) https://www.repository.cam.ac.uk/handle/1810/273560
The aim of this study was to analyse whether, after receiving e-health or e-skills training, the professional applied it to their work, both for their own performance and for the benefit of patients. That's the most important part: does what I learn ultimately benefit the patient? Some references to reinforce this have been introduced in the journal.
3) In accessing the professional’s digital competencies, did you control their previous experience on digital training? Differences in absorptive capability matter. If you were not able to obtain this info, this might be useful in the future study.
In this respect, the participants were asked whether they had attended similar training before. 73 people indicated that this was their first course on digital competence or eHealth. However, as you indicate in your comment, it would be very interesting to find out whether they already have experience with other courses on other topics in digital environments.
4) Lines 288-290: This data could reaffirm what other studies suggested; nurses and younger healthcare professionals use fewer digital resources with their patients p-[26.27]. -> What is the rationale behind this? What do the existing study say?
According to these studies, this could be due, in the case of nurses, to a generation gap or a lack of training received by younger nurses during their undergraduate studies. However, as they themselves point out, more research is needed to be able to conclude the causes of this age disparity. Additional sentences regarding this topic have been introduced in the paper about this topic.
5) Figure 5: Why do we care about the number of resources created? The more the better?
We welcome your comments on this issue. At the time of the questionnaire we thought it might be useful to ask whether they had done different resources but obviously, as you point out, we cannot say that doing three is better than doing one. We have decided to remove this question as it does not provide relevant information.
6) Similar to the comment 5, what do we care about different type of resources that have been developed? Theoretically, or at least practically, what is the key difference among blogs, podcasts, videos, infographics, profiles, etc.? You should have at least provided the evidence that some type is more efficient in communicating with patients or transferring knowledge.
The aim here was not really to analyse whether the resources developed were better or worse, but to find out what practitioners did with what they learned. Your question would be analysed in a second stage with a larger volume of subjects. Knowing what they find easier or more useful can be a way of establishing new proposals for future training. We conducted a review on the use of video to support patient education and found it to be useful. As indicated in the text, there is also evidence on the use of infographics to educate patients. This study did not aim to compare the effectiveness of the different resources but only to explore the current reality.
7) Isn’t there any need to differ the transfer to the workplace and patients? I believe the content will be different.
In reality, digital competence is a macro-competence which, in turn, is made up of other smaller competences. These include information, communication, safety, security, creation, etc. A course designed to consolidate these competences and make them transferable into practice should ideally cover all Kirkpatrick levels. These competences would include those that make me more competent for myself (information search, diary management, document management, etc.), those that help me to relate to others (communication) and those that enable me to be able to create resources for others (videos, web pages, etc.).
Minor issues
1) The limited institutional support and low training professionals receive make this gap even more complex between the popular demand and what healthcare workers offer [17]. -> This sentence can be just added to the previous paragraph, not requiring a separate paragraph. Changes made following your suggestion.
2) Nevertheless, and even though parameters such as satisfaction (level 1) and knowledge (level 2) are often used to assess training suitability, some authors point out that the knowledge and skills learned by nurses are usually transferred to their practice [26]. -> Don’t quite understand why this is problematic. You said, transferring of knowledge is important, right? This needs more explanation.
It is not problematic; quite the contrary. Some authors point out that even though nurses receive training that covers only the two initial Kirkpatrick levels, they transfer what they learn into practice. We qualify the phrase to make it easier to be understood.
3) Line 237: six teen -> sixteen Change made. Thanks for detecting the mistake
4) Line 239: tick -> pick Change made. Thanks for detecting the mistake
5) According to the professional profiles there are only 5 doctors out of 104 respondents, and most of them are nurses. This can be problematic as people generally think of doctors when they say health professionals. That said, isn’t it the case that the program itself is more useful for nurses than doctors?
An explanation has been added in the text to clarify this in order to avoid any possible confusion.

Reviewer 4 Report
Globalization and quick technological advances, force health professionals to develop new competencies to adapt to the new ICT challenges. However, necessary skills such as using digital tools are primarily ignored by institutions, hospitals, universities, etc., forcing professionals to undertake training in these areas independently.
Therefore the objective of this study was to analyze, utilizing consultation with healthcare professionals (HCPs), if there is a certain transfer of what has been learned in the digital healthcare field to the professional practice of HCPs and patients. It was also of interest to analyze whether there are differences between certain professional categories and other characteristic features (age, workplace etc.)?
The manuscript is well written and has a clear friendly structure (Introduction, Materials and Methods, Results, Discussion, Limitations and Conclusions). The subject is interesting and valuable, as the paper raises the important issue of transferring digital competencies of HCPs to their workplace. The background is transparent and informative. The research group is relatively small and the description of the methodology and results needs to be supplemented (see below). The discussion is sufficient and is supplemented with limitation section. The paper is rich in 38 adequate references. The text is complemented by two tables and five figures.
Major points
1. The research methodology should be described in more detail. I cannot find any statistical analysis in the results section. Tables could be used for this purpose.
Minor points
2. The research group is not homogeneous in terms of age and profession. Thus, the generalization of the results should be formulated with caution. This should be emphasized more clearly in the limitation.
Author Response
REVISOR 4
Globalization and quick technological advances, force health professionals to develop new competencies to adapt to the new ICT challenges. However, necessary skills such as using digital tools are primarily ignored by institutions, hospitals, universities, etc., forcing professionals to undertake training in these areas independently.
Therefore the objective of this study was to analyze, utilizing consultation with healthcare professionals (HCPs), if there is a certain transfer of what has been learned in the digital healthcare field to the professional practice of HCPs and patients. It was also of interest to analyze whether there are differences between certain professional categories and other characteristic features (age, workplace etc.)?
The manuscript is well written and has a clear friendly structure (Introduction, Materials and Methods, Results, Discussion, Limitations and Conclusions). The subject is interesting and valuable, as the paper raises the important issue of transferring digital competencies of HCPs to their workplace. The background is transparent and informative. The research group is relatively small and the description of the methodology and results needs to be supplemented (see below). The discussion is sufficient and is supplemented with limitation section. The paper is rich in 38 adequate references. The text is complemented by two tables and five figures.
Major points
- The research methodology should be described in more detail. I cannot find any statistical analysis in the results section. Tables could be used for this purpose.
We have added the p-value in the description of the corresponding results (we prefer not to summarize them in a table since it will require a strong reformatting of the paper). To make clear, we emphasize when there is a strong statistical significance; for example, see the new paragraph in the revised manuscript:
We found a strong dependency (p=0.0238) between the duration of the course and the use of it: the longer the training course, the more it is used; for example, while no one attending to short-term courses (less than 22 hours) use it every day, the percentage increases to 10.8% for mid-term courses and to 42.8% for long-term courses (more than 150 hours).
Also notice that as we mention in the title of the paper this is a pilot study so we are more interested in finding the relevant aspects relating digital training competencies and how they are used (if that is the case) in real world practice than in a deep statistical analysis of any of these features. Future research should include a more complex and specific study of the relationships discovered in this work with a proper statistical design.
Minor points
- The research group is not homogeneous in terms of age and profession. Thus, the generalization of the results should be formulated with caution. This should be emphasized more clearly in the limitation.
The sentence in the limitations section is modified to clarify the impossibility of extrapolating the results, just creating the ground for a larger and deeper analysis on this topic.

Round 2
Reviewer 2 Report
感谢作者对我的评论的详细回复和回答。
Reviewer 4 Report
I am fully satisfied with your corrections and additions.
I have no more comments.